# LLM-Assisted Fast and Customized Model Generation: A Preliminary Exploration

## Abstract

The rapid advancement of AI models has significantly impacted daily life, with Large Language Models (LLMs) playing a pivotal role in automating tasks and providing all-in-one solutions via API services. Meanwhile, there is a growing demand for private, resource-constrained, customizable, and high-performance models tailored to specific user needs. However, many users struggle to deploy these models due to limited resources or technical expertise. In this work, we try to address these challenges by focusing on two primary objectives: (1) to meet the specific needs of a broad range of users, and (2) to lower the barriers to AI model usage (*e.g.*, resource constraints, technical expertise) for most users. In our preliminary exploration, we introduce FLAME, a framework that determines and generates AI models based on data or task descriptions provided by users. While existing solutions rely on pre-built models or extensive finetuning, FLAME leverages LLMs (*e.g.*, GPT4-turbo) to capture data patterns and task features from user input, converting them into user requirements and structured metadata (*e.g.*, task type, model architecture, and classifier dimension). Then, FLAME uses them as guidance to generate customized models by hypernetworks. This approach significantly improves efficiency, achieving up to 270x faster model production compared to finetuning-based paradigms (e.g., all-parameter and LoRA fine-tuning) while maintaining comparable performance across various tasks. We validate the effectiveness of FLAME through comprehensive experiments on Natural Language Processing (NLP), Computer Vision (CV), and tabular datasets, demonstrating its ability to quickly deliver high-quality, customized models.

## 1 Introduction

Recent advancements in AI models, especially in LLMs such as GPT-4 (OpenAI, 2023), LLaMA 3.1 (AI, 2024) and Phi-3 (Abdin et al., 2024) have significantly influenced our daily life (Zhao et al., 2023; Yang et al., 2023). Leveraging their impressive abilities, LLMs offer an all-in-one solution for versatile user requirements through API services, making advanced AI accessible for tasks like text generation, summarization, and chatbots. However, there is still a growing demand for private (deployed), customizable, and resource-constrained models suited to specific domains (Staab et al., 2023; Yao et al., 2023). Since user requirements vary widely, deploying general models might not always achieve optimal results, particularly in specialized areas such as law, economics, and medicine. In contrast, tailored models tend to exhibit superior performance (Turc et al., 2019; Gunasekar et al., 2023; Fu et al., 2023; Hsieh et al., 2023; Yao et al., 2024; Chen & Varoquaux, 2024). However, users might not have adequate expertise or enough data, time and resources to determine the model and finetune it. These barriers greatly hinder the wider application of AI models. Therefore, our research aims **(I) to meet the specific needs of a broad range of users** and **(II) to lower the barriers to AI model usage (*e.g.*, resource constraints, technical expertise) for most users.**

However, these pursuits face certain challenges. First, **changes in user requirements can lead to model adjustments at varying levels**. While minor changes in user requirements may necessitate adjusting the parameters of the target model for better performance (Sagawa et al., 2020; Lv et al., 2023), significant task alterations (*e.g.* regression to classification, data modality change) might require the model to change its output dimensions or even its architecture. Second, **to reduce constraints on AI model usage, both general capabilities (*e.g.*, task understanding) and precise**

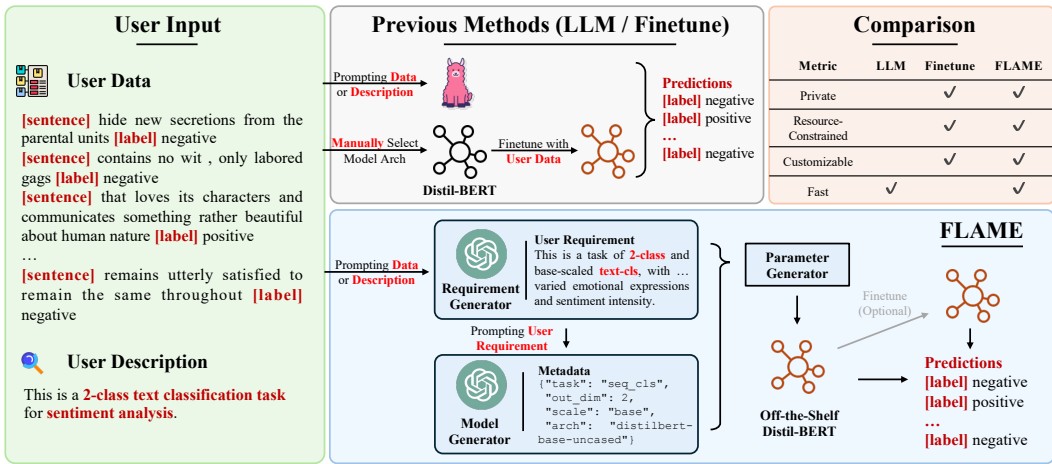

Figure 1: Overview of the framework of FLAME and comparison with previous paradigms.

**customization, ideally without extensive training, are essential.** In real-world scenarios, data often is limited or lacks insufficient supervisions (Wang et al., 2021), and sometimes only a basic task description is provided. Without strong task understanding, accurately capturing user needs becomes difficult, and without precise customization, those needs may not be fully met. Third, **finding efficient methods to provide optimal models, without requiring extensive resources or expertise, remains a key challenge.**

To address the above challenges, we humbly think that **one could leverage the complementary strengths of general large models and specific small models**. Specifically, we could unleash the capabilities of large models to capture user requirements and use them to generate customized small models.

In our preliminary exploration, we initiate our idea into FLAME, as depicted in Figure 1. Specifically, given users' input (User Data, User Description, or both), FLAME constructs prompts to utilize LLMs (GPT4-turbo) to summarize the task, analyze data patterns and task features, and format it into a user requirement (just a single sentence) and structured metadata (determines the most appropriate target model for the given task). Next, FLAME uses Multi-Head Module-Wise Parameter Generator to decode user requirements into model parameters to output the tailored model, which could be directly used by users for prediction. For instance, as shown in Figure 1, for a text classification task of sentiment analysis, users can provide data batches (User Data) or just describe the task (User Description). FLAME then constructs a prompt, interacts with LLM, and outputs User Requirement in Requirement Generator and structured Metadata in Model Generator. Next, we use User Requirement and Metadata to generate the model parameters by Parameter Generator. An optional finetune process (either full-parameter or LoRA (Hu et al., 2022) finetuning) can be undergone for better performance. Finally, users can apply this tailored model to their data. In short, our contributions can be summarized as follows:

- We propose a novel framework FLAME to determine and generate AI models tailored to user data or task description effectively and efficiently.

- FLAME involves Multi-Head Module-Wise Parameter Generator for adjustable and task-conditioned parameters, which extends LoRA-based hypernetworks to more model architectures.

- We conduct extensive experiments in NLP, CV, and tabular data. FLAME can generate tailored models at most 270x faster than previous methods, while still maintaining comparable performance.

## 2 RELATED WORKS

### 2.1 LARGE LANGUAGE MODELS

In recent times, the field of natural language processing (NLP) has been significantly reshaped by the emergence of large language models (LLMs) like ChatGPT (Wang et al., 2019a), GPT-4 (OpenAI, 2023), LLaMA (Touvron et al., 2023), and others. The concept of LLM arises from language

model (Vaswani et al., 2017; Devlin et al., 2019), an algorithm used in natural language processing to predict the likelihood of a sequence of words occurring in a sentence. Characterized by deep architectures, billions of parameters, and tremendous training corpus, LLMs have drastically enhanced the ability of machines to understand, interpret, and generate human language (Naveed et al., 2023; Brown et al., 2020).

Upon their introduction, LLMs have quickly gained widespread attention and have been applied across various domains, including machine translation, text completion, conversational agents, and so on (Shen et al., 2023; Romera-Paredes et al., 2023; Pan, 2023). However, despite their impressive capabilities, LLMs come with their own set of challenges. Research indicates that in certain specific areas, smaller models can outperform LLMs (Turc et al., 2019; Gunasekar et al., 2023; Fu et al., 2023). Moreover, due to the immense size and complexity of these models, they are often impractical for users to employ or fine-tune, particularly when faced with limitations in computational resources or technical expertise (Hu et al., 2022).

## 2.2 HYPERNETWORKS

Hypernetwork, the model designed to output the weights of another model is first proposed by Ha et al. (2017). Since it only needs a single forward pass to output model parameters, it provides a fast and efficient alternative to the vanilla pretrain-finetune paradigm. Given its unique capability, it has gained wide attention in various fields like recommendation system, natural language processing, and computer vision. Lv et al. (2023) proposes a framework DUET for efficient device model generalization, which uses hypernetwork to generate the MLP layers of device models for model personalization. Alaluf et al. (2022) proposes HyperStyle, which learns to modulate StyleGAN's weights to faithfully express a given image in editable regions of the latent space. Ivison et al. (2023) proposes HINT, which uses hypernetwork to encode task definitions into task-conditioned LoRA adapters (Hu et al., 2022) and applies them to LLMs. To clarify, it's important to note that **HINT and FLAME differ greatly in both motivations and technical details**. HINT focuses on using hypernetworks to make instruction tuning more efficient for LLMs, but ours is broader. We aim to generate a variety of models that meet specific user needs, which involves significant technical differences from HINT. For more information, we kindly refer readers to Section 3.

While most hypernetworks are mlps, recently, a few works discuss the potential of hypernetworks with more complex architectures, like GAN (Ratzlaff & Li, 2019), ResNet (Alaluf et al., 2022). These works explore the potential of hypernetworks in model generation to some extent.

## 3 METHODOLOGY

The workflow of FLAME consists of 2 main modules: Requirement Generator and Model Customizer, as depicted in Figure 2. Requirement Generator takes User Data or User Description as input and outputs User Requirement, while Model Customizer translates User Requirement into an off-the-shelf AI model. In this section, we elaborate on the details of our framework.

### 3.1 REQUIREMENT GENERATOR

Given User Data or User Description, Requirement Generator interacts with LLM (GPT4-turbo) to analyze data patterns and task features and summarize them into one sentence: User Requirement $r \in R$.

Effective prompt design is crucial for accurately distilling patterns from data. On the one hand, User Data might be insufficient, and the lack of data poses challenges to reflect the real distribution in users' scenarios. On the other hand, LLMs tend to highlight simpler patterns directly inferable from labels. User Description could enable LLM to focus more precisely on the proper and unique patterns. Generally, we summarize the demands of prompt design as follows:

- The metadata of the task (*e.g.* task type, output dimensions) must be pointed out in the final sentence.

- Data-specific information, if any, should be reflected in the final sentence and must only focus on the data itself rather than the labels given.

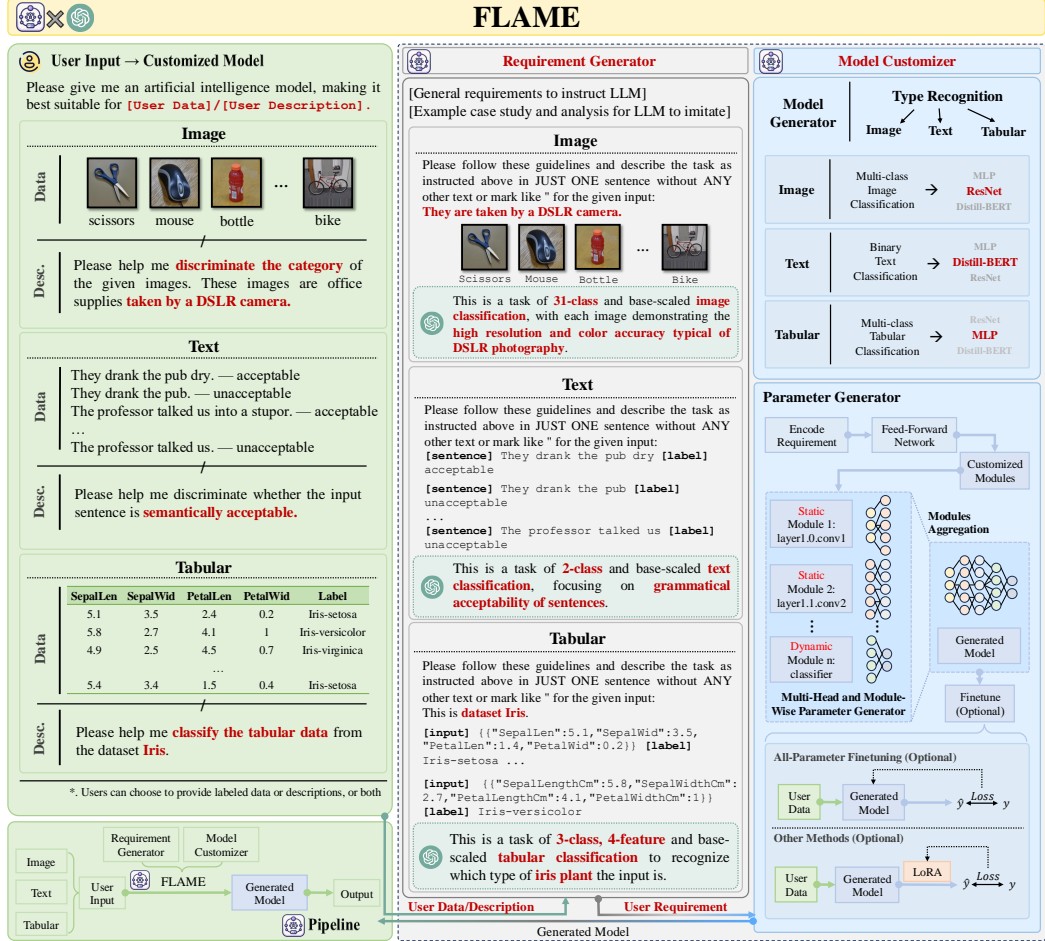

Figure 2: Details of the workflow of FLAME.

The first one is easy to understand, since accurately capturing the task's nature, such as task type and model dimension, is essential for identifying an appropriate model. In contrast, the second one is less intuitive. While a general model may perform adequately in standard scenarios, it often struggles with special data patterns like domain shifts in specific user contexts, leading to significant performance drops (Wang & Deng, 2018; Zhou et al., 2022). Consequently, Requirement Generator must detect these data patterns present in the data, like spurious correlations between background elements and labels for precise customization. The examples of User Requirements can be seen in Figure 2. The prompt of Requirement Generator can be seen in Appendix C.

## 3.2 MODEL CUSTOMIZER

Given User Requirement, Model Customizer translates it into a tailored model. It is comprised of 2 sub-modules, Model Generator and Parameter Generator, responsible for architecture and parameter generation individually.

### 3.2.1 MODEL GENERATOR

Given User Requirement, Model Generator determines the architecture of the target model by prompting GPT4-turbo. The output, denoted as Metadata, is a json with keys: task, out_dim, scale, arch and in_dim. task represents the type of the task (*e.g.*, img_cls) and arch means the architecture of the output model. in_dim and out_dim represents the number of input (used for tabular tasks only) and output features of the task. scale determines the scale of the output model (*e.g.*, DistilBERT-base or DistilBERT-large) and is selected according to users' resources. task, scale and arch has pre-defined choices. For simplicity, we fix scale to be base and leave more options for future work. The prompt of Model Generator can be seen in Appendix D.

### 3.2.2 PARAMETER GENERATOR

Once target model architecture $T$ is determined, Parameter Generator $P(\cdot; \theta_p = (\theta_e, \theta_g))$ generates the parameters $\theta_t$ with User Requirement $r \in R$ as input, following an encoder-decoder architecture, where the encoder $E(\cdot; \theta_e)$ encodes $r$ into a latent variable, the decoder (Multi-Head and Module-Wise Parameter Generator) $G(\cdot; \theta_g)$ decodes it into model parameters module by module.

The Encoder $E(\cdot; \theta_e) : R \mapsto \mathbb{R}^d$ is a language model (DistilBERT-base) followed by a feed forward network for further feature transformation. Following previous solutions (Wang et al., 2019b) to get the latent variable of User Requirement $r$, before FFN, we use the latent of the first token `[CLS]` as the sentence embedding, formatted as Equation (1):

$$z = E(r; \theta_e). \tag{1}$$

Vanilla hypernetworks then decode $z$ into the modules of $\theta_t$ (e.g., `layer1.0.weight`) one by one. However, the diversity of User Requirements brings **three challenges**.

**The diversity of User Requirements leads to variations in architecture even for similar tasks.** A subtle difference in requirements can significantly alter the target model architecture, making it challenging for hypernetworks. For instance, switching from MRPC to STS-B in the GLUE Benchmark (both tasks involve semantic similarity) changes the model requirement from a 2-class classification to regression. However, hypernetworks could not generate these different architectures simultaneously and efficiently. Our preliminary solution is to use a **multi-head** approach. We first **split** the target model modules into two types: the **static components** (e.g., inner layers, remains consistent across tasks), and the **dynamic components** (e.g., the final classifier, varies across tasks). Hypernetworks can easily generate the static component, as its structure remains unchanged across user requirements. For the dynamic component, which poses more challenges, we assign a decoder head to each task. This ensures that the output shape in each head is fixed, allowing the model to efficiently adapt to changes in architecture.

For instance, to generate a DistilBERT-base model for both MRPC and STS-B, we need to divide its modules into static components (*e.g.*, `layer1.0.weight`) and dynamic components (*e.g.*, `classifier.weight` is $768 \times 2$ for MRPC and $768 \times 1$ for STS-B). During inference, we first output the parameters of the static components. Then, for MRPC, to handle the dynamic component, we switch to the MRPC-specific decoder head, outputting modules like $768 \times 2$ `classifier.weight`. For STS-B, the only change is switching to the STS-B-specific decoder head. It's important to note that this approach works only for tasks seen during training. For new test tasks, we could only manually select the most similar task head. Addressing this limitation is left for future work.

**Direct generating large models would result in convergence issues.** Generally, it is not practical to output the parameters of large-scale models directly for convergence issues (Dinh et al., 2022; Alaluf et al., 2022). As a result, we add **LoRA adapters** (Hu et al., 2022) into the model, generate their parameters, and finally obtain the target model by merging them.

**While reducing generation size by only producing LoRA adapter weights can cut down on complexity, it doesn't always lead to optimal model performance.** Previous works like HINT (Ivison et al., 2023) have demonstrated that hypernetworks can generate LoRA adapters to adjust transformer models. Yet, this success might not extend to non-transformer models like ResNet, suffering from performance degradation. We've discovered that this issue arises from overlooking the adjustable layers present in these models (*e.g.* norm layers). Unlike the LayerNorm in transformers, other norm layers, such as BatchNorm are learnable. However, outputting their parameters together with LoRA adapters would greatly increase the scale of hypernetworks, resulting in convergence issues or out-of-CUDA-memory errors. Therefore, we disable the functionality of these layers. For implementation details, we kindly refer readers to Appendix B.

Solving the above challenges, we generate the parameters $\theta_t$ and aggregate them into the model $T(\cdot; \theta_t)$ as shown in Equation (2).

$$\theta_t = G(z; \theta_g). \tag{2}$$

The training process of this module is demonstrated in Algorithm 1. To train Parameter Generator, we provide sufficient task-requirement pairs $A = \{(D_i = \{X_i, Y_i\}, r_i)\}_{i=1}^{N}$. Given a certain requirement $r_i$, we follow the above procedure to obtain the parameter $\theta_{ti} = P(r_i; \theta_p)$ for the target

---

**Algorithm 1** Pseudo-code of Parameter Generator $P(\cdot; \theta_p)$

---

**Require:** $A = \{(D_i = \{X_i, Y_i\}, r_i)\}_{i=1}^N$
**Ensure:** $\theta_p = (\theta_e, \theta_g)$ satisfies Equation (4)
  $i \leftarrow 1$
  **for** $\_ = 0$ to $\sharp$epoch **do**
    **for** $(D_i, r_i)$ in $A$ **do**
      **for** batch in $D_i$ **do**
        Obtain $\theta_t$ with Equations (1) and (2)
        Use batch to compute the loss, update $\theta_t$ and get the update difference $\Delta\theta_t$
        Use $\Delta\theta_t$ to compute the gradients of $\theta_p$ and update $\theta_p$
      **end for**
    **end for**
    Save best checkpoint according to Equation (4)
  **end for**

---

model $T$. Using model $T$ and data $D_i = \{X_i, Y_i\}$, we compute the loss based on the task type, as shown in Equation (3). The loss function $l$ is chosen according to the task: Cross-Entropy for classification tasks, and MSE for regression tasks. We then update $\theta_t$ based on the loss. Finally, the difference in $\theta_t$ before and after the update is used to compute the gradients of $\theta_p$ and update $\theta_p$.

$$L_i = \mathbb{E}_{(x,y)\sim D_i} l(T(x; \theta_{t_i}), y). \tag{3}$$

It is important to note that, during inference, we only need the user requirement $r$ to infer the parameters, which is the key factor behind the speedup compared to other paradigms.

To obtain the best performance on all task-requirement pairs, we require the average loss to be minimal, as shown in Equation (4). Generally, the number of data samples varies from task to task, which could result in an uneven number of task-requirement pairs being generated and in turn lead to inadequate training for tasks with fewer data samples. Such an imbalance is detrimental to the model's overall performance. To address this issue, we manually adjust the portions of each task during the construction of task-requirement pairs. The portion could be found in Appendix A.

$$\hat{\theta}_p = \arg\min_{\theta_p} \frac{1}{N} \sum_{i=1}^N L_i. \tag{4}$$

## 4 EXPERIMENTS

To evaluate our framework, we analyze FLAME with the following questions:

1. **Can FLAME be effectively and efficiently applied to different modalities?**
2. **Can FLAME generalize to unseen tasks while maintaining performance and efficiency?**
3. **How well does FLAME's output model serve as a foundation for further adaptations?**

### 4.1 EXPERIMENT SETTINGS

Three settings are selected for our main experiments: NLP, CV, and tabular data.

**NLP.** We use GLUE Benchmark (Wang et al., 2019b), which has nine sentence- or sentence-pair language understanding tasks built on established existing datasets and selected to cover a diverse range of dataset sizes, text genres, and degrees of difficulty[1]. **Distil-BERT base** is the target model.

**Tabular Data.** We choose 10 famous tabular classification tasks from UCI Machine Learning Repository[2]: Iris (Unwin & Kleinman, 2021), Heart Disease (Detrano et al., 1989), Wine (Aeberhard et al., 1994), Adult (Becker & Kohavi, 1996), Breast Cancer (Street et al., 1993), Car Evaluation (Bohanec & Rajkovic, 1988), Wine Quality (Cortez et al., 2009), Dry Bean (Koklu & Özkan, 2020), Rice (Cınar & Koklu, 2019), Bank Marketing (Moro et al., 2014). **MLP** is the target model.

---

[1]https://gluebenchmark.com/
[2]https://archive.ics.uci.edu/

---

**CV.** We use the Office-31 dataset (Saenko et al., 2010), which is commonly used in domain adaptation, to evaluate **both the effectiveness, efficiency and zero-shot ability** of our approach. This dataset contains 31 object categories in three domains: Amazon, DSLR, and Webcam with 2817, 498, and 795 images respectively, different in background, viewpoint, color, etc. **ResNet-50** (He et al., 2016) is the target model. In the main experiment, we first train our model with Amazon and DSLR, directly feed User Requirements extracted by LLM on Webcam's training data to FLAME, and test the output model on Webcam's test set, where FLAME sees no Webcam's data but its requirements. We have more zero-shot experiments in Section 4.3.1.

### 4.1.1 BASELINES

Generally speaking, FLAME introduces a novel capability: translating user data or descriptions into model parameters. As this is the first framework of its kind, there are no directly comparable baselines. To address this, we compare FLAME with two widely adopted training paradigms: **Finetune** and **LoRA**, which are standard for adapting models to new tasks. In Finetune, we finetune the target model with all parameters by the training data **in each task individually**. Since **FLAME uses LoRA to reduce the complexity**, we treat finetuning the target model with LoRA adapters **in each task** as the baseline LoRA. The LoRA adapters are the same as FLAME's. In tabular experiment, since MLP is simple, FLAME directly outputs its weights, rather than using LoRA adapters. Mind that **in each setting, we ONLY need ONE FLAME to solve the tasks**, while other paradigms need finetuning for each tasks.

For our study, we introduce a variant of FLAME, **FLAME-F**. This adaptation includes an additional step where, following FLAME's generation, we perform full-parameter finetuning using consistent hyperparameter settings (1 epoch if no specified). Details are available in Appendix A.

### 4.1.2 METRICS

In addition to the performance metric, we stand at the viewpoint of common users WITHOUT technical expertise and propose additional metrics. We humbly think that **it is NOT how long it takes to train a FLAME but how long they could get a model that counts for the most users**. Hence, we evaluate FLAME with two additional efficiency metrics: **E2E Runtime** and **Relative Efficiency**. E2E (end-to-end) Runtime measures total task completion time (seconds), while Relative Efficiency scales this runtime against the worst-performing method.

### 4.2 RESULTS AND OBSERVATIONS

The results can be seen from Tables 1 to 3, underscoring the efficiency, and satisfactory performance of our framework. **For hyperparameter settings, we kindly refer readers to Appendix A.** Here, we provide detailed discussions of our results.

**FLAME yields progressively more significant speed gains as the size of the target model increases.** Leveraging the power of hypernetworks, FLAME generates custom model weights in a single forward pass, eliminating the need for a resource-intensive and expertise-dependent finetuning process. This approach yields progressively more significant speed gains over the conventional pretrain-finetune paradigm as the size of the target model increases. The acceleration observed ranges from approximately 40x for a simple MLP (1K) to 270x for Distil-BERT base (66M), marking a 7-fold increase in efficiency. It is also interesting to find out that in the experiments of tabular data, LoRA is a bit slower than Finetune. The deficiency in speed stems from the target model. Since tabular tasks are rather simple, FLAME directly uses MLP, a very shallow neural network isomorphic to LoRA adapters. Hence, directly finetuning the target model is more efficient than using LoRA to finetune it. These results highlight FLAME's exceptional efficacy in producing tailored models, particularly for larger target architectures.

**Inter-task knowledge empowers FLAME for enhanced model generation.** Due to hypernetworks' limitations, FLAME cannot generate large models directly. Instead, our implementation for sizable target models is to generate LoRA adapters and merge them to construct the final models. This approach may initially seem at most comparable to the baseline LoRA. Yet, in practice, FLAME surpasses LoRA in all experiments and even outperforms Finetune in CV and tabular data tasks. This performance boost is largely attributable to the inter-task knowledge gleaned by FLAME.

Table 1: Detailed results on GLUE with Distil-BERT as the target model. We use GLUE's metrics to evaluate these tasks. ♯Epoch represents target model's training epochs for each method to obtain the results. E2E (end-to-end) Runtime measures total task completion time (seconds), while Relative Efficiency scales this runtime against the worst-performing method.

| | | | | | | | | | | | | | Results on GLUE Benchmark (Distil-BERT) | | |
|---|---|---|---|---|---|---|---|---|---|---|---|---|---|---|---|
| Methods | CoLA | SST-2 | MRPC | STS-B | QQP | MNLI-m | MNLI-mm | QNLI | RTE | WNLI | DM | Score | ♯Epoch | E2E Runtime (s) | Relative Efficiency |
| LoRA | **48.3** | 91.0 | 84.9 / 80.3 | 81.2 / 80.0 | 68.9 / 87.3 | 80.5 | 33.1 | **88.1** | 52.8 | **65.1** | 0.0 | 71.5 | 20 | 75672 | 1.3 |
| Finetune | 45.5 | **91.3** | **86.6 / 80.8** | **82.1 / 80.9** | 69.2 / 87.8 | **81.8** | **80.8** | 87.6 | 56.9 | 63.7 | **35.6** | **74.4** | 20 | 95870 | 1.0 |
| FLAME | 39.5 | 88.9 | 85.3 / 78.4 | 80.9 / 80.3 | 63.3 / 83.5 | 77.8 | 78.0 | 84.6 | 69.5 | 64.4 | 28.0 | 73.4 | **0** | **350** | **273.8** |
| FLAME-F | 36.9 | 90.8 | 85.5 / 79.4 | 81.3 / 80.5 | 67.0 / 86.6 | 77.8 | 78.1 | 85.8 | **70.0** | 62.3 | 29.9 | 73.8 | 1 | 1101 | 87.0 |

Table 2: Detailed results of various methods on 10 tabular classification tasks with accuracy as the evaluation metric. ♯Epoch represents target model's training epochs for each method to obtain the results. E2E (end-to-end) Runtime measures total task completion time (seconds), while Relative Efficiency scales this runtime against the worst-performing method.

| | | | | | | | | | | | Results on Tabular Data (MLP) | | | |
|---|---|---|---|---|---|---|---|---|---|---|---|---|---|---|
| Methods | Iris | Heart Disease | Wine | Adult | Breast Cancer | Car Evaluation | Wine Quality | Dry Bean | Rice | Bank Marketing | Average | ♯Epoch | E2E Runtime (s) | Relative Efficiency |
| LoRA | 93.3 | **63.0** | 67.3 | 54.7 | 95.9 | 71.3 | 55.0 | **88.9** | 92.5 | 89.8 | 77.2 | 20 | 272 | 1.0 |
| Finetune | 88.9 | 54.3 | 89.1 | **55.2** | **96.5** | 71.0 | 55.3 | **90.6** | **93.1** | 89.9 | 78.4 | 20 | 233 | 1.2 |
| FLAME | **100.0** | 60.9 | **94.5** | 54.7 | 95.3 | **71.5** | 54.1 | 85.0 | 92.5 | 89.8 | 79.8 | **0** | **6** | **46.2** |
| FLAME-F | **100.0** | 62.0 | **94.5** | 55.1 | 95.9 | 71.3 | 55.4 | 88.8 | 92.9 | **90.0** | **80.6** | 1 | 14 | 20.2 |

Table 3: Detailed results of various methods on Office-31 (31-class classification). The metric is accuracy, top-3 & 5 accuracy. Our results on Webcam are conducted with no training data provided. ♯Epoch represents target model's training epochs for each method to obtain the results. E2E (end-to-end) Runtime measures total task completion time (seconds), while Relative Efficiency scales this runtime against the worst-performing method. The average only considers Amazon and DSLR here.

| | | | | | | | Results on Office-31 (ResNet-50, FLAME is ZERO-SHOT in Webcam) | | | | | | | |
|---|---|---|---|---|---|---|---|---|---|---|---|---|---|---|
| Domain | Amazon | | | DSLR | | | Average | | | Webcam | | | | | |
| Methods | Acc | Acc@3 | Acc@5 | Acc | Acc@3 | Acc@5 | Acc | Acc@3 | Acc@5 | Acc | Acc@3 | Acc@5 | ♯Epoch | E2E Runtime (s) | Relative Efficiency |
| LoRA | 66.4 | 77.7 | 84.8 | 78.4 | 92.2 | 96.1 | 72.4 | 85.0 | 90.5 | 72.5 | 87.5 | 93.8 | 400 | 3393 | 1.1 |
| Finetune | 67.5 | 79.2 | 83.7 | 84.3 | 98.0 | 100.0 | 75.9 | 88.6 | 91.9 | 90.0 | 100.0 | 100.0 | 400 | 3770 | 1.0 |
| FLAME | 66.4 | 79.9 | 83.7 | 92.2 | 100.0 | 100.0 | 79.3 | 90.0 | 91.9 | 76.2 | 87.5 | 91.2 | **0** | **15** | **257.6** |
| FLAME-F | 67.8 | 81.3 | 85.9 | 92.2 | 100.0 | 100.0 | 80.0 | 90.7 | 92.8 | 77.5 | 90.0 | 91.3 | 1 | 18 | 206.4 |

Although tasks within a single experiment differ, they share common knowledge. For example, in NLP experiments, both MRPC and QQP tasks focus on semantic equivalence between sentences, and in CV experiments, all domains involve similar classification tasks with unique data-specific characteristics. This observation is also confirmed by our zero-shot success in the Webcam task, where it outperforms LoRA without direct data access, relying solely on User Requirements. We provide further zero-shot analyses in Section 4.3.1

**FLAME not only generates well-performed models but also provides efficient initial weights.** In our main experiments, we introduce FLAME-F which undergoes a 1-epoch full-parameter fine-tuning post-generation. This approach leads to more favorable outcomes, achieving an average performance improvement of 0.8 absolutely while only doubling the total time consumption. We provide detailed analyses of this observation in Section 4.3.2.

## 4.3 FURTHER ANALYSES

To further answer Question 2&3 in Section 4, we conduct further experiments to evaluate FLAME's zero-shot ability (Section 4.3.1) and the capability of weight initialization (Section 4.3.2). In addition, we have also in depth analyzed FLAME's prompt design and case study of Requirement Generator and Model Generator, the impact of user input and the robustness of the quality of User Requirements. We kindly refer readers to Appendices C to F for detailed results.

### 4.3.1 ZERO-SHOT ABILITY

In our main experiments, we evaluated the zero-shot capability of FLAME using the Webcam domain. This section expands the analysis by considering DSLR and Amazon as zero-shot domains,

Table 4: Analyses on the zero-shot ability of FLAME on Office-31. LoRA and Finetune use training data, while Zero-Shot and FLAME see no data in the zero-shot domain. FLAME-F additionally finetunes the target model with zero-shot domain's training data individually.

| Setting | AD → W | | | | DW → A | | | | AW → D | | | |
|---|---|---|---|---|---|---|---|---|---|---|---|---|
| Metrics | A | D | Average | W | D | W | Average | A | A | W | Average | D |
| LoRA | 66.4 | 78.4 | 72.4 | 72.5 | 78.4 | 72.5 | 75.5 | 66.4 | 66.4 | 72.5 | 69.5 | _78.4_ |
| Finetune | 67.5 | 84.3 | 75.9 | _90.0_ | 84.3 | 90.0 | 87.2 | _67.5_ | _67.5_ | **90.0** | **78.8** | **84.3** |
| Is Seen Task? | ✓ | ✓ | ✓ | ✗ | ✓ | ✓ | ✓ | ✗ | ✓ | ✓ | ✓ | ✗ |
| Zero-Shot | _67.7_ | 86.0 | 76.9 | 63.7 | 90.0 | _91.3_ | 90.7 | 17.0 | 67.0 | 85.0 | 76.0 | 70.0 |
| FLAME | 66.4 | 92.2 | _79.3_ | 76.2 | **100.0** | 88.7 | _94.4_ | 19.8 | 64.3 | 83.8 | 74.1 | **84.3** |
| FLAME-F | **67.8** | 92.2 | **80.0** | **91.3** | _98.0_ | 93.8 | **95.9** | **68.2** | 67.8 | _87.5_ | _77.7_ | 84.3 |

training on the remaining two domains separately. This results in three settings: AD→W (Amazon, DSLR → Webcam), DW→A, and AW→D.

Additionally, we performed zero-shot evaluations in NLP using five Natural Language Inference (NLI) tasks: ANLI_R1, ANLI_R2, ANLI_R3 (Liu et al., 2020), CB[3], and MNLI[4], with the last two tasks as zero-shot domains. All of these tasks are to decide the relationship between the premise and the hypothesis (entailment, contradiction, or neutral).

**Since these tasks have the same output dimensions**, to facilitate a more direct comparison, we introduce a new baseline, **Zero-Shot**. For CV, it trains a 31-class ResNet-50 model on two domains and evaluates it on the zero-shot domain. For NLP, it trains a 3-class DistilBERT-base model on ANLI_R1,2 and 3 (abbreviated as R1, R2, and R3) and evaluates it on the zero-shot tasks (CB & MNLI). In contrast, FLAME generates the target model based solely on User Requirements, while FLAME-F further refines it by full-parameter finetuning after FLAME's generation (10 epochs for AD→W, 15 epochs for DW→A and 1 epoch for others).

Table 5: Accuracy on NLI tasks. CB & MNLI are zero-shot. MNLI has 2 sub-tasks.

| Methods | R1 | R2 | R3 | Average | CB | MNLI | Average |
|---|---|---|---|---|---|---|---|
| LoRA | 39.0 | 38.5 | 42.5 | 40.0 | 50.0 | 77.8/78.7 | 68.8 |
| Finetune | 42.0 | **44.7** | **45.3** | 44.0 | _57.1_ | _79.6/79.6_ | _72.1_ |
| | | Seen Tasks | | | | Unseen Tasks | |
| Zero-Shot | 46.8 | _43.4_ | _42.9_ | 44.4 | 28.6 | 57.4/59.4 | 48.5 |
| FLAME | _51.0_ | 41.0 | 41.9 | _44.6_ | 53.6 | 57.8/59.5 | 57.0 |
| FLAME-F | **51.9** | 40.2 | 42.1 | **44.7** | **66.1** | **79.4/80.5** | **75.3** |

The results in Tables 4 and 5 showcase the strong zero-shot capabilities of FLAME, with the model even outperforming LoRA in certain cases. This can be attributed to the combination of **user requirement supervision** and **the inter-task knowledge**. This observation is consistent with the main experiments. The only difference between Zero-Shot and Finetune is that Zero-Shot trains the seen tasks altogether while Finetune trains them separately. **Leveraging inter-task knowledge, Zero-Shot consistently outperforms Finetune across all seen tasks**. Moreover, with User Requirement (have additional knowledge) as supervision, FLAME shows even further improvements over Zero-Shot in some settings, despite being trained with LoRA. Notably, in DW → A, FLAME experiences a sharp performance drop compared to LoRA and Finetune. This decline is attributed to the distinctive nature of Amazon, which exhibits a larger disparity with other domains. Zero-Shot's similar performance in this scenario can support our viewpoint. When FLAME's output undergoes further finetuning for a limited duration, the model shows improved performance, we will in depth analyze this in Section 4.3.2.

### 4.3.2 CAPABILITY OF WEIGHT INITIALIZATION

In Section 4.2, we state that FLAME not only generates well-performed models but also provides efficient initial weights. To further testify to this viewpoint, we finetune the target model (ResNet-50) with all parameters under the same hyperparameter setting to Finetune, with the **zero-shot output** of FLAME on Webcam as weight initialization. It is important to notice that the major difference between the two methods lies in the weight initialization. While Finetune uses the weights pretrained on ImageNet (Deng et al., 2009), ours uses the weights outputted by FLAME in a zero-shot manner.

---

[3]https://super.gluebenchmark.com/

[4]https://gluebenchmark.com/

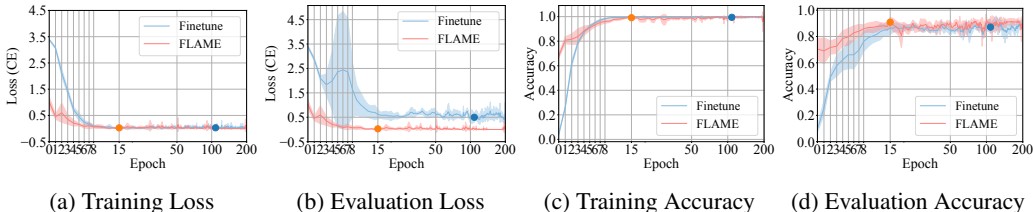

| (a) Training Loss | (b) Evaluation Loss | (c) Training Accuracy | (d) Evaluation Accuracy |

Figure 3: Detailed analyses on the capability of weight initialization of FLAME. For clearer comparison, we increase the length of the starting epochs. Meanwhile, we mark the best checkpoint of each method in the figures with a solid round point.

We save the best checkpoint in evaluation and test it on Webcam's test data. The results, detailed in Table 6, reveal a notable aspect: despite a roughly 10% performance gap compared to Finetune without access to Webcam's data shown in Table 3, our framework exhibits remarkable convergence speed when finetuned with Webcam's training data, using the same hyperparameters as Finetune. Specifically, while Finetune requires 108 epochs to reach optimal evaluation results, our framework, given FLAME's zero-shot output as initialization, achieves comparable performance in just 16 epochs, a 6.75-fold increase in speed.

Table 6: Results on the test dataset using the best evaluation checkpoint of each method. ♯epoch implies the number of epochs for each method to achieve the checkpoint.

| Office-31 (Webcam) Results with Different Weight Initializations | | | |
|---|---|---|---|
| Methods \ Metrics | Acc | Acc@3 | Acc@5 | ♯Epoch |
| Finetune | 90.0 | 100.0 | 100.0 | 108 |
| FLAME | 95.0 | 98.8 | 100.0 | **16** |

Moreover, we meticulously track the progression of training and evaluation losses, alongside the corresponding accuracy as presented in Figure 3 with five different seeds. The depicted curves represent the mean value, while the shaded areas denote the range within one standard deviation. All the figures demonstrate the superiority of FLAME's output as a weight initialization. As shown in Figure 3b, FLAME's initialization outperforms the baseline in evaluation throughout the process. Notably, the substantial standard deviation observed in the baseline during the initial epochs in Figure 3b can be attributed to the instability often encountered at the onset of training. Moreover, while the baseline shows a marginally improved performance in the later stages of training in Figure 3c, our approach demonstrates better performance on evaluation data in Figure 3d, suggesting a better generalization capability and robustness.

## 5 FUTURE WORK AND CONCLUSION

In this work, we introduce FLAME, a framework that leverages LLMs to determine and generate customized models based on user data or task descriptions. While FLAME shows strong performance in generating customized models, efficiently addresses specific user needs and lowers the barrier to AI model usage, it is important to note that this is merely a preliminary exploration, and several challenges remain unsolved.

First, the granularity of Model Generator can be improved. A more detailed analysis of factors such as task complexity and available user resources could enable more refined model architecture decisions. Second, the capabilities of Parameter Generator need expansion. The current multi-head solution is limited to tasks resembling the training data. For tasks with greater disparity (e.g., new output dimensions or modalities), FLAME still falls short. Further, for different modalities, separate FLAMEs are required. Developing an all-in-one FLAME for different modalities is a key goal for future research.

Generally speaking, by introducing FLAME, we aim to paves the way for a new paradigm in adaptive, efficient model creation. However, our research is still in its early stages, and we welcome discussions and collaborative efforts to further explore this emerging field.

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

## A  HYPERPARAMETER SETTING

In the main text, we conduct comprehensive experiments on FLAME. In this section, we provide detailed hyperparameter settings to reproduce our results in Tables 1 to 3, as shown in Table 7. Since FLAME is a framework that generates target models directly. During the training of FLAME (denoted as pretrain for simplicity), we need to train both FLAME and the target model to update the overall framework. Therefore, we set their learning rate and weight decay individually.

Table 7: Detailed Hyperparameter Setting of Our Main Experiments

| Parameter \ Setting | NLP | CV | Tabular Data |
|---|---|---|---|
| GPU | A100 80G | A100 80G | A100 80G |
| Optimizer (FLAME) | Adam | Adam | Adam |
| Learning Rate (FLAME) | 1e-5 | 1e-4 | 1e-3 |
| Weight Decay (FLAME) | 1e-4 | 1e-5 | 1e-4 |
| Optimizer (Target Model) | Adam | Adam | Adam |
| Learning Rate (Target Model) | 1e-4 | 1e-3 | 2e-2 |
| Weight Decay (Target Model) | 1e-4 | 1e-3 | 1e-4 |
| lora_r | 16 | 8 | NA |
| lora_alpha | 32 | 16 | NA |
| lora_dropout | 0.05 | 0.1 | NA |
| target_modules | .*[qv]_lin | layer.\..\.conv. | NA |
| ♯Epoch (Pretrain) | 50 | 100 | 80 |
| Batch Size | 256 | 256 | 64 |
| Latent Dimension | 768 | 128 | 25 |
| Seed | 2024 | 2024 | 2024 |

The hyperparameter settings of baselines are similar. We set the number of training epochs to 20, 200, and 20 in NLP, CV, and tabular data individually with the learning rate to be 1e-3, 1e-3, and 2e-2 individually. The LoRA config of baseline LoRA is the same to FLAME, except that in tabular data, lora_r is 4, lora_alpha is 8, lora_dropout is 0.1, and target modules is mlp\.\d\.*.

Since the portion of each task's training data is imbalanced, we manually re-balance the weight of each task by simply retrain the samples several times. In CV experiments, we do not use this technique. In NLP experiments, we upweight WNLI and RTE with factor 5 and STS-B, CoLA and MRPC with factor 3. In tabular experiments, we upweight Wine, Iris with factor 10 and HeartDisease, Rice and DryBean with factor 2.

## B  IMPLEMENTATION DETAILS

As we stated in the main text, to solve convergence issues, we adopt LoRA adapters to the target model and generate their parameters. Besides, since we only need very simple AI models like MLPs to solve tabular tasks, FLAME directly outputs target models' parameters rather than LoRA adapters' parameters. The implementation of MLP is shown below. We accordingly mark their hyperparameters concerning LoRA to NA in Table 7.

```
class MLP(nn.Module):
    def __init__(self,
                 in_dim: int,
                 out_dim: int,
                 hidden_dim: int,
                 n_layers: int, *args, **kwargs):
        super().__init__(*args, **kwargs)
        self.mlp = nn.Sequential(
            nn.Linear(in_dim, hidden_dim), *[
                nn.Linear(hidden_dim, hidden_dim)
```

```
            for _ in range(n_layers)
        ],
        nn.Linear(hidden_dim, out_dim)
    )

def forward(self, x: Tensor) -> Tensor:
    return self.mlp(x)
```

Generally, in the context of fine-tuning, we could use LoRA adapters to reduce the overall cost. Although such paradigm operates flawlessly in traditional scenarios, it does have some problems in FLAME. To be specific, complex models have parameters that are not trained but changed during the stage of finetuning (*e.g.* running mean and variance of BatchNorm Layers). Due to convergence and CUDA memory consumption issues, it is not practical for us to generate these parameters alongside the generation of LoRA adapters. However, leaving these modules unsettled would result in unacceptable performance degradation. Hence, to enable FLAME to generate complex models, we disable the functionality of these layers in the target model at the expense of less stability with the code below:

```
def train(self, mode=True):
    type(model).train.__call__(self, mode)
    for m in self.modules():
        if isinstance(m, nn.BatchNorm2d):
            m.eval()
            m.weight.requires_grad = False
            m.bias.requires_grad = False

model.train = functools.partial(train, model)
```

## C  PROMPT DESIGN AND CASE STUDY ON REQUIREMENT GENERATOR

Since FLAME directly uses LLM to summarize User Requirements, it is crucial to design proper prompts for them. As discussed in Section 3.1, the prompt should tell the type of the task and the data-specific information. However, due to various reasons (*e.g.* lack of data), summarizing users' requirements often poses challenges. In response, we carefully design the prompt and incorporate users' knowledge (User Data or Description) into it, resulting in better performance.

As shown in Figure 4, the first row is the system prompt (prompt template), which contains our requirement of generating User Requirements, two examples for better reasoning, and the place (last 2 lines) to fill User Description and User Data individually. In the requirement part, we emphasize that LLM ought to succinctly identify the type of the task (*e.g.* classification) and point out the data-specific information. For further uses, we require the output User Requirement must have 5 elements, namely task type, data pattern, number of classes, number of input features (optional) and scale. These features are curial for further process in Model Generator, which determines the architecture of target model. In the example part, we provide 2 examples and analyze them, following the idea of COT (Chain of Thought) (Wei et al., 2022) for better performance.

As shown in Figure 4, in NLP, we use benchmark GLUE for experiments. The major difference between GLUE's tasks can be directly summarized from their task name since they vary in task type (*e.g.* binary two-input classification, multi-class one-input classification, one-input regression.) Therefore, to instruct LLMs to precisely capture user requirements in NLP, we don't need to provide User Description. In the example, with the prompt given, LLM successfully points out that the given data is a task of binary sentiment analysis. Similar results can be observed in tabular data. Here, different from NLP, we also provide background information (User Description) on the given data. We directly tell LLM that the data comes from datasets such as Iris and Car Evaluation. With User Description and User Data provided, LLM successfully points out that the input is a tabular classification to evaluate the acceptability of car purchases. For CV, since the task is multi-class image classification and there is 31 categories, for better accuracy, we directly tell the number of classes in User Description. Without it, LLM would tell that it is multi-class, losing the information

| | |
|---|---|
| **System Prompt** | Suppose you are given either a User Description or a batch of User Data. User Description gives the background information about the task, while User Data could be in any modality such as text, images, tabular data or others and is paired with its label.
Your task is to succinctly identify the type of processing task demonstrated (e.g., classification, detection) based on the information provided.
Focus specifically on the unique characteristics or patterns relevant to User Data or User Description, i.e. presented in the data or described in the text.
The output MUST be ONE sentence, including the following information:
1. task type: a 2-word phrase indicating the data type (like image, text, sequence-to-sequence, tabular, audio) and the task type (like classification, generation, regression, detection) individually.
2. data pattern: the common of User Data (like photography features), MUST unique to this batch and irrelevant to the label, preferable patterns inferable from User Description yet not identical to it.
3. number of classes: specified if the task involves some form of classification (cls).
4. number of input features: specified only in tabular tasks
5. scale: selected from "small", "base", and "large" according to user resources, which, if not noted in advance, should default to "base". |

Table 8: Human-evaluated Average Rank of User Requirements on Office-31.

| Average Rank (↓) of User Requirements on Office-31 (CV exps) | | | |
|---|---|---|---|
| **Settings** | **FLAME** | **w/o User Data** | **w/o User Description** |
| **Amazon** | **1.2** | 2.08 | 2.72 |
| **DSLR** | **1.3** | 2.0 | 2.7 |
| **Webcam** | **1.22** | 2.78 | 2.0 |
| **Average** | **1.24** | 2.29 | 2.47 |

To conduct a more detailed analysis of the impact of User Descriptions on hard tasks, we perform a human-evaluated experiment on Office-31 (CV). Annotators are asked to rank requirements generated by (1) **FLAME**, (2) **w/o User Data**, and (3) **w/o User Description** based on the ground truth provided (50 samples per domain). The results are shown in Table 8. FLAME, when both User Data and User Descriptions are provided, produces the most accurate User Requirements. The average ranking of User Requirements using only User Descriptions is lower than those generated using only User Data, demonstrating the importance of User Descriptions in accurately capturing requirements for difficult tasks.

We attribute this phenomenon to two factors. First, compared to User Data, User Descriptions provide more straightforward supervision, which LLMs can more easily capture and translate into User Requirements. Second, LLMs (GPT-4 turbo in our implementation) still struggle with solving multi-modal tasks, and this limitation might also negatively impact their performance in this context.

# D PROMPT DESIGN AND CASE STUDY ON MODEL GENERATOR

Given User Requirements, Model Generator prompts LLM to get json-format Metadata to determine the architecture of the target model. Examples can be found in Figure 5. Here, LLM is asked to determine the architecture of the target model based on the User Requirement given and pre-defined choices. All modalities use the same prompt.

| | |
|---|---|
| **System Prompt** | Suppose you are asked to specify the desired model architecture information given a User Requirement, which is a sentence describing the user's task. You should detail the following features in a JSON format:

1. task: like img_cls, s2s_gen, seq_cls, tabular_cls, seq_reg.
2. out_dim: a number indicating the output dimension, specified if the task involves some form of classification (cls).
3. in_dim: a number indicating the number of input features, specified only in tabular tasks.
4. scale: selected from "small", "base", and "large" according to user resources, which, if not noted in advance, should default to "base".
5. arch: suggested model architecture, selected from "mlp_small", "mlp_base", "mlp_large", "mobilenet_v3_small", "resnet50", "resnet152", "distilbert-base-uncased", "bert-base-uncased"
As a result, one desired output could be: {"task":"img_cls", "out_dim": 31, "scale": "base", "arch": "resnet50"}

Please provide JUST the JSON output based on the User Requirement given: |
| **NLP Example** | This is a task of **2-class** and **base-scaled text classification**, with each sentence demonstrating varying degrees of **emotional sentiment**. | `{`
`  "task": "seq_cls",`
`  "out_dim": 2,`
`  "scale": "base",`
`  "arch": "distilbert-base-uncased"`
`}` |
| **Tabular Example** | This is a task of **4-class, 6-feature and base-scaled tabular classification** to predict **car evaluation status** based on attributes like buying price and safety. | `{`
`  "task": "tabular_cls",`
`  "out_dim": 4,`
`  "in_dim": 6,`
`  "scale": "base",`
`  "arch": "mlp_base"`
`}` |
| **CV Example** | This is a task of **31-class and base-scaled image classification**, with each image characterized by **varying lighting and clarity typical of webcam photography**. | `{`
`  "task": "img_cls",`
`  "out_dim": 31,`
`  "scale": "base",`
`  "arch": "resnet50"`
`}` |

Figure 5: Prompt Details and Case Study on Model Generator. The prompt remains the same on NLP, CV and tabular modalities and is used to GPT4-turbo to get Metadata. **The green color texts** are those reflecting the correct data-specific information.

Table 9: Detailed results of the influence of the number of User Requirements on the final performance in tabular tasks.

| | | | | | | | | | | | |
|---|---|---|---|---|---|---|---|---|---|---|---|
| **Results on Tabular Data (MLP)** | | | | | | | | | | | |
| ♯Requirements | Iris | Heart Disease | Wine | Adult | Breast Cancer | Car Evaluation | Wine Quality | Dry Bean | Rice | Bank Marketing | Average |
| 1 | 97.8 | 60.4 | 57.4 | 81.2 | 94.7 | 69.0 | 43.4 | 88.0 | 92.3 | 89.8 | 77.4 |
| 2 | 86.7 | 56.0 | 90.7 | **82.2** | 81.3 | 69.0 | 52.0 | **89.0** | 91.5 | 89.8 | 78.8 |
| 5 | **100.0** | **60.9** | **94.5** | 54.7 | 95.3 | **71.5** | **54.1** | 85.0 | **92.5** | 89.8 | **79.8** |
| 10 | 97.8 | 57.1 | 85.2 | 82.0 | **97.7** | 69.0 | 48.1 | 72.6 | 91.8 | **90.0** | 79.1 |
| 20 | 97.8 | 56.0 | 79.6 | 76.4 | 84.2 | 69.4 | 52.9 | 82.8 | 88.8 | 89.8 | 77.8 |

Table 10: Analyses on the zero-shot ability of FLAME on Office-31. LoRA and Finetune use training data, while Zero-Shot and FLAME see no data in the zero-shot domain. FLAME-F additionally finetunes the target model with zero-shot domain's training data for **10 epochs**.

| | | | | | | | | | | | | |
|---|---|---|---|---|---|---|---|---|---|---|---|---|
| **Results on Office-31 (ResNet-50, FLAME is ZERO-SHOT in Webcam)** | | | | | | | | | | | | |
| **Domain** | | Amazon | | | DSLR | | | Average | | | Webcam | |
| **Methods** | Acc | Acc@3 | Acc@5 | Acc | Acc@3 | Acc@5 | Acc | Acc@3 | Acc@5 | Acc | Acc@3 | Acc@5 |
| LoRA | 66.4 | 77.7 | 84.8 | 78.4 | 92.2 | 96.1 | 72.4 | 85.0 | 90.5 | 72.5 | 87.5 | 93.8 |
| Finetune | 67.5 | 79.2 | 83.7 | 84.3 | 98.0 | 100.0 | 75.9 | 88.6 | 91.9 | 90.0 | 100.0 | 100.0 |
| Is Seen Task ? | | ✓ | | | ✓ | | | ✓ | | | ✗ | |
| Zero-Shot | 67.7 | 80.5 | **86.2** | 86.0 | 98.0 | 100.0 | 76.9 | 89.3 | **93.1** | 63.7 | 78.8 | 83.8 |
| FLAME | 66.4 | 79.9 | 83.7 | **92.2** | 100.0 | 100.0 | 79.3 | 90.0 | 91.9 | 76.2 | 87.5 | 91.2 |
| FLAME-F | **67.8** | **81.3** | 85.9 | **92.2** | 100.0 | 100.0 | **80.0** | **90.7** | 92.8 | **91.3** | 100.0 | 100.0 |

Table 11: Analyses on the zero-shot ability of FLAME on Office-31. LoRA and Finetune use training data, while Zero-Shot and FLAME see no data in the zero-shot domain. FLAME-F additionally finetunes the target model with zero-shot domain's training data for **15 epochs**.

| | | | | | | | | | | | | |
|---|---|---|---|---|---|---|---|---|---|---|---|---|
| **Results on Office-31 (ResNet-50, FLAME is ZERO-SHOT in Amazon)** | | | | | | | | | | | | |
| **Domain** | | DSLR | | | Webcam | | | Average | | | Amazon | |
| **Methods** | Acc | Acc@3 | Acc@5 | Acc | Acc@3 | Acc@5 | Acc | Acc@3 | Acc@5 | Acc | Acc@3 | Acc@5 |
| LoRA | 78.4 | 92.2 | 96.1 | 72.5 | 87.5 | 93.8 | 75.5 | 89.9 | 95.0 | 66.4 | 77.7 | **84.8** |
| Finetune | 84.3 | 98.0 | 100.0 | 90.0 | 100.0 | 100.0 | 87.2 | 99.0 | 100.0 | 67.5 | **79.2** | 83.7 |
| Is Seen Task ? | | ✓ | | | ✓ | | | ✓ | | | ✗ | |
| Zero-Shot | 90.0 | 100.0 | 100.0 | 91.3 | 98.8 | 100.0 | 90.7 | **99.4** | 100.0 | 17.0 | 29.8 | 39.4 |
| FLAME | 100.0 | 100.0 | 100.0 | 88.7 | 98.8 | 100.0 | 94.4 | 99.4 | 100.0 | 19.8 | 33.6 | 45.2 |
| FLAME-F | 98.0 | 100.0 | 100.0 | 93.8 | 98.8 | 100.0 | **95.9** | 99.4 | 100.0 | **68.2** | 78.4 | 83.7 |

Table 12: Analyses on the zero-shot ability of FLAME on Office-31. LoRA and Finetune use training data, while Zero-Shot and FLAME see no data in the zero-shot domain. FLAME-F additionally finetunes the target model with zero-shot domain's training data for **1 epoch**.

| | | | | | | | | | | | | |
|---|---|---|---|---|---|---|---|---|---|---|---|---|
| **Results on Office-31 (ResNet-50, FLAME is ZERO-SHOT in DSLR)** | | | | | | | | | | | | |
| **Domain** | | Amazon | | | Webcam | | | Average | | | DSLR | |
| **Methods** | Acc | Acc@3 | Acc@5 | Acc | Acc@3 | Acc@5 | Acc | Acc@3 | Acc@5 | Acc | Acc@3 | Acc@5 |
| LoRA | 66.4 | 77.7 | **84.8** | 72.5 | 87.5 | 93.8 | 69.5 | 82.6 | 89.3 | 78.4 | 92.2 | 96.1 |
| Finetune | 67.5 | 79.2 | 83.7 | **90.0** | 100.0 | 100.0 | **78.8** | **89.6** | 91.9 | 84.3 | 98.0 | 100.0 |
| Is Seen Task ? | | ✓ | | | ✓ | | | ✓ | | | ✗ | |
| Zero-Shot | 67.0 | 77.7 | **84.8** | 85.0 | 95.0 | 97.5 | 76.0 | 86.4 | 91.2 | 70.0 | 82.0 | 86.0 |
| FLAME | 64.3 | 78.8 | 83.7 | 83.8 | 97.5 | 97.5 | 74.1 | 88.2 | 90.6 | 84.3 | 96.1 | 96.1 |
| FLAME-F | **67.8** | **80.2** | 84.1 | 87.5 | 96.3 | 97.5 | 77.7 | 88.3 | 90.8 | **96.1** | 98.0 | 98.0 |

# E    ROBUSTNESS OF MODEL CUSTOMIZER

As introduced in Section 3.2, Model Customizer is trained with requirement-data pairs and optimized for the given batch of data. The pairs are created randomly from the Cartesian product of the requirement set and dataset to ensure Model Customizer's robustness. We evaluate how the number

of requirements affects results in Tabular experiments. Results can be found in Table 9. As shown in the results, the number of User Requirements has a certain impact on the final performance. However, the influence is not significant, demonstrating Model Customizer's stability across the size of requirement set. Meanwhile, we can conclude from the results that to obtain optimal performances, a medium number (roughly 5) of requirements would be better.

# F  DETAILED RESULTS ON FLAMES'S ZERO-SHOT ABILITIES

In Section 4.3.1, we analyze the zero-shot ability of FLAME on Office-31. Due to space reason, we only demonstrate the Accuracy metric, we put full results in Tables 10 to 12 for reference.

