# OpenReview forum: "LLM-Assisted Fast and Customized Model Generation: A Preliminary Exploration"
_ICLR.cc/2025/Conference — Submitted to ICLR 2025_

### Official Review · Reviewer_c6Wh · 2024-10-28

**Soundness:** 2
**Presentation:** 3
**Contribution:** 3
**Rating:** 5
**Confidence:** 4

**Summary:**

This paper introduces FLAME, a framework that leverages LLMs such as GPT4-turbo to generate customized AI models based on user input, including data or task descriptions. FLAME processes user input to generate prompts that utilize LLMs to summarize the task, analyze data patterns and task features, and convert this information into a one-sentence user requirement and the target model’s structured metadata. The framework then uses a LoRA-assisted Parameter Generator to transform the user requirement into model parameters, producing a tailored model. The authors claim that FLAME can generate models up to 270 times faster than traditional fine-tuning methods while maintaining competitive performance. The core innovation lies in the use of hypernetworks to generate model parameters, focusing on reducing computational costs and complexity. The paper demonstrates FLAME’s applicability across NLP, CV, and tabular data tasks.

**Strengths:**

S1: The framework is well-structured, clearly written, and provides complete code. Within the same modality, FLAME can address different AI tasks, contributing to the potential advancement toward AGI.

S2: In terms of methodology, the Requirement Generator is key to understanding user needs, with carefully designed prompts that consider both task metadata features and data-specific patterns. The Parameter Generator incorporates LoRA adapters to reduce the number of generated model parameters, while avoiding convergence and CUDA memory consumption issues by disabling some functionality in adjustable layers of complex models.

S3: The experimental section is extensive, covering tasks across three modalities (language, tabular data, and image), with evaluations based on three well-chosen metrics (performance, end-to-end runtime, and relative efficiency). The paper also provides an in-depth analysis of FLAME’s zero-shot capability, weight initialization, and includes essential details on prompt design, case studies, and robustness assessments.

**Weaknesses:**

W1: The Model Generator section presents pre-defined choices for tasks, scales, and architectures, especially with limited architecture options per modality (as shown in Appendix D, there are fewer than three models for each task). This raises concerns about FLAME’s effectiveness and generalization across more complex tasks, particularly considering that FLAME’s goal is to generate customized small models.

W2: The experiments lack important baseline comparisons. Although the authors claim in Section 4.1.1 that FLAME is the first framework to translate user data or descriptions into model parameters, it would be valuable to include comparisons with state-of-the-art LLM-based frameworks that also generate AI/ML solutions from user requirements, such as AutoM3L [1] and AutoMMLab [2]. Additionally, comparisons with traditional AutoML systems are missing, which would provide a more thorough evaluation of FLAME’s performance.

W3: The experiments only evaluate GPT4-turbo’s performance on the benchmarks, which leaves open the question of whether the proposed method relies heavily on GPT4-turbo’s capabilities. This raises concerns about the technical soundness and generalizability of the approach when applied to other LLMs.

W4: One of the significant concerns with integrating LLMs into AI solution generation is the risk of data leakage. Since LLMs are pre-trained on large amounts of publicly available data, including many common ML datasets, this overlap could lead to biased evaluations [3]. The paper does not address how this potential issue is mitigated.

Minor Comments:

The motivation for using hypernetworks to generate model parameters in this work needs further explanation.


[1] Luo D, Feng C, Nong Y, et al. AutoM3L: An Automated Multimodal Machine Learning Framework with Large Language Models[J]. arXiv preprint arXiv:2408.00665, 2024.

[2] Yang Z, Zeng W, Jin S, et al. AutoMMLab: Automatically Generating Deployable Models from Language Instructions for Computer Vision Tasks[J]. arXiv preprint arXiv:2402.15351, 2024.

[3] Jeong D P, Lipton Z C, Ravikumar P. Llm-select: Feature selection with large language models[J]. arXiv preprint arXiv:2407.02694, 2024.

**Questions:**

Please see the Weakness and Minor Comments above.

---

### Official Review · Reviewer_cAMM · 2024-10-29

**Soundness:** 2
**Presentation:** 3
**Contribution:** 1
**Rating:** 3
**Confidence:** 3

**Summary:**

The authors propose a method for taking a user-generated description of a machine learning task and (optionally) a small set of input/output data examples and turning them into an adequately selected deep learning model initialized with weights that allow it to perform the given task. In addition, it is possible to fine-tune the outputted model for one epoch to maximize performance. The proposed method consists of two main steps: (1) prompt an LLM to interpret the user input and convert it into a single-sentence requirement summary, and (2) take the requirement in order to (a) prompt an LLM to turn it into a JSON-formatted specification of the model architecture, and (b) generate the parameters using a trained model based on HyperNetworks. The main benefit put forth by the authors is the ability to generate models with performance comparable to fine-tuning pre-trained models at a fraction of the cost. This benefit stems from the ability of the framework to generate a trained model using a single forward pass, followed by an optional single epoch of fine-tuning.

**Strengths:**

**(S1)** From a user experience perspective, the proposed framework aims to produce a fully trained model with minimal effort on the part of the user, which is quite a compelling use case.

**(S2)** The paper is relatively well-written and easy to follow. The authors make an effort to highlight important statements and guide the overall flow of the paper.

**(S3)** The ability to produce trained models with such a low compute cost is also pretty compelling.

**Weaknesses:**

**(W1)** Although the authors do not explicitly state this in the paper (perhaps intentionally?), this work firmly overlaps with the established areas of meta-learning and transfer learning. As such, the key challenge here is generalizability to unseen datasets. I was not able to find details about which datasets were used for training the FLAME framework and which ones were used for inference. Without this, it becomes hard to judge how well the proposed method can generalize to datasets it never encountered. The authors were pretty open about this being a shortcoming of the current approach (e.g. in Section 3.2.1 and Section 5) which is commendable. However, I would argue that generalizability is actually both the hardest and the most interesting part. Furthermore, without it, the entire framework can be seen as an interesting idea without much real-world use.

**(W2)** Leaning on the above point, I think this paper should be placed in the context of other work related to meta-learning and transfer learning, both in terms of methodology (see W1) and in terms of updating the introduction and related work sections.

**(W3)** Given that the authors apply HyperNetworks to obtain model weights, it's easy to wonder how much the proposed method relies on the parameter generator simply memorizing the weights for all of the dataset/task pairs it has been trained on, as opposed to being able to acquire meta-knowledge that is transferable between tasks. In other words, do we even need hypernetworks or can we just get away with a collection of pre-trained model weights? There are certain claims made in e.g. Section 4.2 about that topic, but this is more of a post-hoc interpretation rather than an empirically validated claim. More convincing empirical evidence would be to include results where we have a collection of pre-trained models (the same collection as the one FLAME ends up learning to parameterize during pre-training) and test if having an LLM pick the most appropriate set of weights from the stash could improve performance.

**(W4)** (minor issue) There are a small handful of writing issues and missing clarifications that could easily be addressed:
 * (Abstract) "high-performance models", and in general usage of the word "model" is a little ambiguous as it is unclear if the authors are talking about LLMs or deep learning models, or even classical ML models. This becomes clear later on but could have been clarified earlier.
 * (Section 1, page 2) "In real-world scenarios, data often is limited or lacks insufficient supervisions" -- Firstly, "supervisions" -> "supervision". Secondly, the authors likely want to say "lacks sufficient supervision". Thirdly, it is unclear what the authors even mean when they say the data lacks sufficient supervision. This should probably be reworded.
 * (Section 4.1.2) "while Relative Efficiency scales this runtime against the worst-performing method" -- worst-performing in terms of runtime or model quality? I think the authors mean runtime but this could be clarified.
 * (Section 5) "we aim to paves" -> "we aim to pave"

**Questions:**

**(Q1)** Which datasets were used for training FLAME and which were used only for inference? Was FLAME trained only once on all three modalities or was there a different model depending on the modality? Please clarify these details for every single experiment.

**(Q2)** I understand that this might be too much to ask in the rebuttal cycle, but would you be able to provide some convincing evidence that the HyperNetwork is (a) needed (i.e. cannot be replaced by a simple stash of pre-trained models); and (b) is able to adapt the parameters to a specific task that it has not seen before (it would be especially interesting to see how a change in the task requirement impacts the change in model parameters, both for better and for worse)?

---

### Official Review · Reviewer_Fi8D · 2024-10-30

**Soundness:** 3
**Presentation:** 3
**Contribution:** 2
**Rating:** 6
**Confidence:** 3

**Summary:**

The authors introduce a framework named FLAME, which is designed to address users' diverse demands for AI models and lower the barriers to AI model usage.
FLAME does not use a LLM to solve all types of tasks, but utilizes LLM to interpret users' requirements in plaintext into metadata. And the metadata is then used to guide the generation of customized models through hypernetworks.
This method has a great acceleration effect compared to finetuning the model.

**Strengths:**

1. The idea is novel, allowing the large model to generate the small model to complete specific tasks, which is helpful in scenarios where local resources are limited and privacy is a concern.
2. Experiments show that there is a significant performance improvement compared to the finetune method
3. The work is solid, an end-to-end framework is implemented, and source code is provided

**Weaknesses:**

## 1. With the development of LLM, the problems mentioned in the paper can be solved by LLMs themselves in a more efficient way
This paper mainly focuses on three types of data: text, table, and image. Problems with these three types of data can already be solved by multimodal LLM, and there are already some powerful open source multimodal LLMs.
These LLMs can be deployed locally to avoid privacy issues and can be fine-tuned to support customization.
The method proposed in the paper requires (1) using LLM to generate metadata, then (2) using metadata to generate models, and finally (3) using the generated model to solve the problem.
If a multimodel LLM is used, you only need to adjust the prompt in the first step, and the next two steps can be omitted.
It would be better to add a discussion about the trade-offs between using LLM directly and using FLAME.

## 2. The proposed method has limited scope of application
The paper divides the tasks into three categories, and only considers classification and regression tasks. Other types (such as audio) are not supported yet. In addition, this classification by data type will make some tasks involving mixed data of multiple modalities impossible to complete, which will become an obvious limitation of the architecture.

**Questions:**

1. The task-based generation model seems very novel, but how effective is it for a completely new type of task? It would be better to give a few examples.
2. If the task type is very classic, is it necessary to generate a new model, because such tasks may already have more effective models on model hosting sites such as huggingface? Is it possible to integrate FLAME with existing pre-trained models for common tasks?

---

### Official Review · Reviewer_Fowy · 2024-11-02

**Soundness:** 2
**Presentation:** 3
**Contribution:** 2
**Rating:** 3
**Confidence:** 3

**Summary:**

The paper introduces FLAME, a framework that utilizes LLMs and hypernetworks to dynamically generate AI models tailored to specific user needs. It efficiently translates user inputs into structured requirements through prompts and enables rapid production of customized models across domains like NLP, CV, and tabular data, significantly speeding up model generation compared to traditional fine-tuning while maintaining comparable performance.

**Strengths:**

1. Comprehensive Experiments: The paper provides an extensive set of experiments across various domains and perspectives, which strengthens the credibility of the results and supports the claims of the framework's capabilities.
2. Interesting Concept: By leveraging hypernetworks to customize AI models from user inputs, this approach presents a promising and interesting idea that could lead to innovative developments in automated model customization.

**Weaknesses:**

1. Lack of Practicality: The paper presents an ambitious concept of "generating AI models for all users." However, its applicability is limited to only three tasks and models: NLP (Distil-BERT), CV (ResNet-50), and Tabular (MLP). This limitation diminishes its practical significance in real-world applications.
2. Lack of Innovation: The paper primarily focuses on using hypernetworks with traditional models (Distil-BERT 2019, ResNet 2016, and earlier MLPs), where the role of large language models (LLMs) is merely to optimize descriptions through prompting. This limited integration offers little novelty or practical enhancement, particularly in light of recent advancements in the integration of LLMs and hypernetworks, e.g., using hypernetworks with LLMs for domain adaptation [1], and hypernetwork-based multi-objective fine-tuning for LLM alignment [2].
3. Manual Intervention & Generalizability: The use of FLAME highly relies on manual selection of datasets, models, and parameter adjustments. For instance, when addressing new tasks, it requires manually selecting the most similar task head (line 248). Additionally, the weight for each task must be manually set (line 296). Consequently, I am concerned that the strict constraints on corresponding tasks and models may hinder the generalization capability of FLAME.
4. Background and Premise Concerns: The paper's premise that users lack sufficient data, time, and resources (lines 16, 44), which contrasts with its use of small models (max 66M, line 366), questioning the validity of its foundational assumptions about resource constraints.
5. Efficiency Trade-offs: The paper highlights a significant speed increase (270x, line 99), which is an important metric from user's perspective. However, comparing only the inference speed without considering the pretraining costs (Appendix A) could result in a less comprehensive analysis.
---
[1] Hypernetwork-Assisted Parameter-Efficient Fine-Tuning with Meta-Knowledge Distillation for Domain Knowledge Disentanglement

[2] HyperDPO: Hypernetwork-based Multi-Objective Fine-Tuning Framework

**Questions:**

1. As mentioned in lines 247-249 regarding the adaptability issues for new tasks, could the authors explain how FLAME can be applied to model generation for specific new tasks? Additionally, please discuss any ideas for improving FLAME's generalizability to new tasks without manual intervention.
2. How does FLAME address scalability issues, especially when dealing with larger models (e.g. Llama3-8B) or more complex datasets?
3. Given the limited number of tasks and models, what are the differences in performance between using FLAME and a rule-based model selection approach?
4. Although the basic information about the pretraining stage is briefly mentioned in Appendix A, considering the overhead involved, could the authors provide more detailed information on the pretraining costs?

---

### Official Review · Reviewer_jZor · 2024-11-04

**Soundness:** 2
**Presentation:** 2
**Contribution:** 2
**Rating:** 3
**Confidence:** 4

**Summary:**

The paper proposes a method to leverage LLMs to generate ML models based on data and task descriptions provided by users. The purpose is twofold: 1. general purposed models do not perform well for specific user tasks; 2. there is a barrier for ordinary users to build models customised to their tasks.

**Strengths:**

Structuralising the way users leverage existing ML models for their own tasks is useful. The paper explores the topic of understanding user requirements for dynamic model generation.

**Weaknesses:**

1. The objective of the paper is to make a layman capable of generating a ready-to-use model to analyse their own data. With a layman assumption, the simple data and task description input may be ambiguous and not have an one-to-one mapping to the target model. This may happen when the requirement is underspecified as the user lacks of knowledge about ML models and the terminology to describe their data and tasks. The proposed method does not seem to discuss the complexity in requirement generation.

2. The paper argues its difference to HINT is that the proposed method applies broadly to different networks other than those used in NLP. However, The HINT approach does not have to be limited to NLP models. It aims to generate model parameters to attach to a pre-trained model. In this sense, the novelty of the proposed method seems is more on the requirement generation part, which is a bit ad-hoc because of weakness 1. In addition, the prompts for model architecture generation ask LLMs to choose from a limited and pre-defined model architectures before applying a similar approach for parameter generation. This is a trivial extension to HINT.

3. In order to generate parameters for non-transformer models, the paper disables batchnorm and other normalisation layers, which seems to remove the functionalities of functional models simply to fit for the parameter generation purpose. It might not be a good practice for designing a software system.

**Questions:**

LLMs often enable complicate tasks through iterations in chats. How does the proposed method ensure the requirement from a user is fully captured? In addition, how does a layman know if the generated model meets the requirement and what the user may do when it does not?

---

### Meta-Review · Area_Chair_2Jdj · 2024-12-19

**Metareview:**

The paper provides framework named FLAME, which aims to make AI models more accessible to non-experts. The reviews agree the problem is well motivated. Several reviewers also positively mentioned the extensive experiments showing promising results. This being said, the reviews raise several crucial concerns. First is the practicality of the paper, both in terms of the requirements from the user (jZor: “With a layman assumption, the simple data and task description input may be ambiguous ... proposed method does not seem to discuss the complexity in requirement generation.”, Fowy: “the use of FLAME highly relies on manual selection of datasets, models, and parameter adjustments.”) and in terms of the limited scope (Fowy: “... applicability is limited to only three tasks and models:...” c6Wh: “The Model Generator section presents pre-defined choices for tasks, scales, and architectures, especially with limited architecture options per modality”). Another notable issue raised is the need for a better comparison against baselines, either multi-modal LLMs (mentioned by Fi8D) or other frameworks (AutoM3L, AutoMMLab, mentioned by c6Wh).

The raised concerns are too major to allow for the paper to be published in its current form.

**Additional Comments On Reviewer Discussion:**

.

---

### Decision · Program_Chairs · 2025-01-22

Reject